# Comparative Effectiveness of Apixaban and Rivaroxaban Lead-in Dosing in VTE Treatment: Observational Multicenter Real-World Study

**DOI:** 10.3390/jcm12010199

**Published:** 2022-12-27

**Authors:** Omar A. Alshaya, Ghazwa B. Korayem, Majed S. Al Yami, Asma H. Qudayr, Sara Althewaibi, Lolwa Fetyani, Shaden Alshehri, Fai Alnashmi, Maram Albasseet, Lina Alshehri, Lina M. Alhushan, Omar A. Almohammed

**Affiliations:** 1Department of Pharmacy Practice, College of Pharmacy, King Saud bin Abdulaziz University for Health Sciences, Riyadh 14611, Saudi Arabia; 2Pharmaceutical Care Services, King Abdulaziz Medical City, National Guard Health Affairs, Riyadh 11426, Saudi Arabia; 3King Abdullah International Medical Research Center, Riyadh 11481, Saudi Arabia; 4Department of Pharmacy Practice, College of Pharmacy, Princess Nourah bint Abdulrahman University, P.O. Box 84428, Riyadh 11671, Saudi Arabia; 5Department of Clinical Pharmacy, College of Pharmacy, King Saud University, Riyadh 12371, Saudi Arabia; 6Department of Clinical Pharmacy, College of Pharmacy, Taif University, Taif 21944, Saudi Arabia; 7Pharmacoeconomics Research Unit, College of Pharmacy, King Saud University, Riyadh 12371, Saudi Arabia

**Keywords:** venous thromboembolism, bleeding, lead-in, oral anticoagulant, apixaban, rivaroxaban

## Abstract

Apixaban and rivaroxaban require lead-in dosing for 7 and 21 days, respectively, when treating venous thromboembolism (VTE). However, no evidence exists to support subtracting parenteral anticoagulation days from total lead-in dosing. A multicenter study was conducted, including adult patients with acute VTE who received apixaban or rivaroxaban. The patients were grouped as follows. The recommended group received oral lead-in anticoagulant for the full recommended duration. The mixed group received lead-in therapy as parenteral with oral anticoagulant. The incidence of recurrent VTE (rVTE) and major bleeding (MB) within 90 days were the main outcomes. Of the 368 included patients, 47.8% received apixaban, and 52.2% received rivaroxaban. The recommended lead-in was used in 296 patients (80.4%), whereas 72 (19.6%) received the mixed-lead-in regimen. Five patients had rVTE events within 90 days; two occurred during hospitalization in the recommended group versus none in the mixed group (0.7% vs. 0.0%; *p* = 1.000). After discharge, two events occurred in the recommended group and one in the mixed group (0.7% vs. 1.4%; *p* = 0.481). In terms of MB, 24 events occurred in 21 patients within 90 days. During hospitalization, 11 events occurred in the recommended group and seven in the mixed group (3.7% vs. 9.7%; *p* = 0.060). After discharge, five more events occurred in the recommended group and one in the mixed group (1.4% vs. 1.7%; *p* = 1.000). The mixed-lead-in regimen is safe and effective in comparison with the recommended-lead-in regimen.

## 1. Introduction

The risk of recurrent venous thromboembolism (rVTE) is highest during the acute phase or in the first week after the occurrence of a new VTE event. Data pooled from 15 studies in a meta-analysis investigating the time frame posing the highest risk of rVTE found that the first week after the index VTE event was associated with the highest risk of developing rVTE, followed by the second, third, and fourth weeks; at week five, the risk declined and stabilized. Furthermore, the first month after VTE had the highest risk of rVTE in comparison to the second and third months [1]. Moreover, the risk of death was at its highest during the first month after the occurrence of a VTE event [2]. As a result, pivotal clinical trials of approved direct oral anticoagulants (DOAC) for VTE treatment have taken this critical phase into consideration [3].

The phase 2 studies for apixaban and rivaroxaban suggested that initial intensified regimens enhance the effect of the anticoagulant, which can be beneficial in the acute phase of VTE treatment [4,5,6]. In the AMPLIFY and EINSTEIN trials, a monotherapy approach with high lead-in doses (10 mg twice daily (BID) for seven days with apixaban and 15 mg BID for three weeks with rivaroxaban) was considered, followed by maintenance doses for the rest of the treatment duration [7,8,9]. By contrast, the HOKUSAI-VTE and RE-COVER trials were designed by initiating therapy with a parenteral anticoagulant for 5–10 days, followed by switching to oral therapy with edoxaban or dabigatran [10,11].

In phase 3 studies for apixaban and rivaroxaban, most patients received parenteral anticoagulation for less than 48 h prior to initiating DOACs [7,8,9]. However, in real-world practice, patients may stay on parenteral anticoagulation for more than 48 h based on multiple reasons, which may include the need for frequent examinations and the interruption of anticoagulation. Clinicians facing this scenario find it challenging to decide whether to use the lead-in therapy for the full recommended duration. Therefore, some clinicians decide to subtract the number of days for which a patient receives parenteral anticoagulation from the total recommended-lead-in duration for DOACs, while others incorporate the full lead-in duration regardless of initial parenteral therapy.

Thus, a study that included 171 patients was conducted to assess the effectiveness and safety of two dosing strategies: full lead-in therapy or reduced lead-in therapy. Although the researchers found a higher rate of rVTE in the reduced compared to the full lead-in group (5% vs. 1%), the difference was not statistically significant [12]. However, this lack of significance may not accurately reflect the importance of receiving the full lead-in therapy. For example, some patients in the reduced lead-in group did not receive any lead-in therapy, and the small sample size might have hindered the finding of significant results. Moreover, the reduced-lead-in group experienced significantly more bleeding events (16% vs. 2%, *p* = 0.004) [12]. Although this investigation is the only published study to highlight the need to devote more attention to lead-in therapy, patients with <24 h of parenteral anticoagulation were excluded from the study for unclear reasons. 

In light of the scarcity of evidence about this issue, we aimed to assess the effectiveness and safety of two lead-in dosing regimens on the incidence of rVTE, major bleeding (MB), and clinically relevant non-major bleeding (CRNMB) in patients with acute VTE events.

## 2. Materials and Methods

### 2.1. Study Design, Setting, and Patients

A multicenter retrospective cohort study was conducted, including data from January 2016 to December 2020. Patients who had acute deep-vein thrombosis (DVT), pulmonary embolism (PE), or both at King Abdulaziz Medical City (KAMC), King Saud University Medical City (KSUMC), and King Abdullah bin Abdulaziz University Hospital (KAAUH) in Riyadh, Saudi Arabia, were evaluated. Adult patients (≥18 years) diagnosed with a new VTE event who used either apixaban or rivaroxaban for treatment and were not on therapeutic anticoagulants prior to the index VTE event were eligible for the study.

Eligible patients were then evaluated for inclusion in one or the other of the two groups in the study: the recommended-lead-in group and the mixed-lead-in group. The recommended-lead-in regimen was defined as receiving the recommended 7 days of apixaban 10 mg BID or 21 days of rivaroxaban 15 mg BID with no more than 2 days of prior parenteral anticoagulation. The mixed-lead-in regimen was defined as patients who received a parenteral anticoagulant and then apixaban 10 mg BID or rivaroxaban 15 mg BID to complete a total duration of at least 6 or 19 days of combined lead-in therapy, respectively. Patients were excluded if: (1) they were switched to maintenance doses other than the recommended ones (5 mg BID for apixaban and 20 mg daily for rivaroxaban), or (2) they received lead-in therapy for more than the maximum duration of 9 days for apixaban or 23 days for rivaroxaban (i.e., the total duration for parenteral and oral anticoagulant) as followed in the AMPLIFY and EINSTEIN trials [7,8,9]. Parenteral anticoagulants included unfractionated heparin (UFH), low-molecular-weight heparin (i.e., enoxaparin), and fondaparinux. The determination of the parenteral anticoagulant dose was left at the discretion of the treating team. KAMC was the main center for apixaban treatment, while KSUMC and KAAUH comprised the centers where rivaroxaban was administered. The study was approved by the supervising IRB at the three sites; KSUMC (ref. no. E-21-6295), KAAUH (ref. no. 22-0139), and KAMC (ref. no. NRC21R/400/09).

### 2.2. Data Collection and Outcomes

The collected data included patients’ demographics, medical history, VTE characteristics (i.e., type, location, etiology, date of the event, history of previous VTE), VTE risk factors (i.e., history of major or orthopedic surgery within one year, thrombophilia, active cancer, use of oral contraceptive, obesity (body mass index; BMI ≥ 30), and immobility), hemoglobin level, renal function at the time of initiating lead-in anticoagulant therapy (serum creatinine, estimated glomerular filtration rate (eGFR), and creatinine clearance (CrCl) based on the Cockcroft–Gault equation), and the use of antithrombotic agents. Furthermore, data were collected for the type and duration of therapeutic parenteral anticoagulation (if used) and for the lead-in and maintenance regimen for apixaban or rivaroxaban (i.e., dosing for lead-in and maintenance, along with duration of lead-in therapy). The risk of bleeding in the study was estimated using the simplified formula developed by Kuijer et al [13].

The efficacy outcomes were the incidence of rVTE during hospitalization or within 30 or 90 days of the indexed VTE event (i.e., excluding events that occurred during hospitalization). The safety outcomes were the incidence of MB and CRNMB during hospitalization or within 30 or 90 days of the indexed VTE event. Other outcomes included the incidence of rehospitalization for VTE-related causes within 30 or 90 days of the indexed VTE event and all-cause death during hospitalization. The MB and CRNMB were defined based on the International Society on Thrombosis and Haemostasis (ISTH) criteria [14].

### 2.3. Statistical Analysis

Descriptive statistics were used to describe the participants’ characteristics. Unpaired *t*-test and chi-square or Fisher’s exact tests were used for continuous and categorical variables, respectively, to compare and examine differences between the two groups (recommended vs. mixed). The groups were compared to each other in terms of patients’ baseline characteristics, along with the study efficacy and safety outcomes. The sample size was estimated to be 122 patients for an effect size of 0.3 and 80% power of test, as calculated by RStudio software (version: 2022.07.2-576). Comparisons with *p*-value <0.05 were considered statistically significant. Data extraction from electronic health records was conducted using the research electronic data capture (REDcap, Vanderbilt University, Nashville, TN, USA). Data were managed using Microsoft Excel, version 2010 (Microsoft Corp., Redmond, WA, USA), and all statistical analyses were performed using the SAS software, version 9.4 (SAS Institute Inc., Cary, NC, USA).

## 3. Results

### 3.1. Patients’ Demographics and Medical Histories

In all, 698 patients with acute VTE events who received treatment using apixaban or rivaroxaban were screened for inclusion. Out of this number, 368 patients were included; the remaining patients were excluded for the reasons illustrated in Figure 1. The recommended-lead-in group had 296 patients, and the mixed-lead-in group had 72 patients. Overall, the mean age for patients was 53.1 ± 19.6 years; in addition, 64.4% were female, and mean BMI was 30.8 ± 7.0 kg/m^2^. The mean duration for hospital stay was 6.1 ± 13.3 days, with a significantly longer duration of hospitalization for patients in the mixed compared to the recommended-lead-in group (12.2 days vs. 4.6 days; *p* < 0.001). Comorbidities such as atrial fibrillation, coronary artery disease, hypertension, valvular diseases, diabetes mellitus, thrombophilia, and active cancer were non-significantly more prevalent in the mixed-lead-in group. The history of MB was more prevalent in the mixed-lead-in group (11.1% vs. 4.7% for the recommended); meanwhile, the history of CRNMB was more prevalent in the recommended-lead-in group (3.4% vs. 1.4% for the mixed). Notably, these differences were not statistically significant, as can be seen in Table 1.

For the risk factors for rVTE, history of previous VTE was more prevalent—but not significant—in the recommended-lead-in group (11.8% vs. 5.6% for the mixed). Notably, most of these historical VTE events occurred >12 months before the current event in both groups. The rates of immobility (25% vs. 22%) and history of major general (15.3% vs. 10.1%) or orthopedic surgery (11.1% vs. 7.1%) were all higher in the mixed compared to the recommended-lead-in group. Note that most of the major or orthopedic surgeries were within the three months preceding the current hospitalization. Table 1 summarizes these data about the risk factors for VTE recurrence.

On the day that the lead-in anticoagulant was initiated, the mean eGFR was higher for the patients in the mixed group compared to the patients in the recommended group (115.4 ± 65.7 vs. 99.6 ± 40.5 mL/min/1.73 m^2^; *p* = 0.009), with no difference between the groups regarding CrCl. Meanwhile, the median level of Hgb before the initiation of lead-in therapy was lower for the patients in the mixed compared to the patients in the recommended-lead-in group (11.7 ± 2.2 vs. 12.7 ± 2.0 g/dL; *p* < 0.001). Table 2 offers laboratory data concerning the renal-function tests and hemoglobin.

### 3.2. VTE Events and Medications Used

The most prevalent type of VTE among the patients in the recommended group was DVT (47.6%), while PE was the most prevalent type in the mixed group (75.0%), and the difference in the types of index VTE events was statistically significant (*p* < 0.001). Among all the index VTE events, 55.2% were provoked, with comparable rates between the two groups. The overall risk of bleeding was comparable between the groups, and most of the patients had low-to-intermediate risk (20.1% and 76.9%, respectively). A detailed comparison of these characteristics can be found in Table 3.

Apixaban was used by 149 patients (50.3%) in the recommended group; meanwhile, rivaroxaban was used by 45 patients (62.5%) in the mixed group. Among the patients in the recommended-lead-in group, 33.1% did not use any parenteral anticoagulant. However, when parenteral anticoagulants were used, LMWH (enoxaparin) was the most commonly used agent (56.8% in the recommended group and ~ 85% in the mixed group). The mean duration of the parenteral anticoagulant use was 1.3 ± 0.4 days, and the mean total duration of the parenteral with the oral anticoagulant was 8.3 ± 0.4 or 22.2 ± 0.4 days for patients on apixaban or rivaroxaban, respectively.

For the patients in the mixed-lead-in group, the mean duration of parenteral anticoagulant use ranged between 2.2 ± 1.4 and 5.6 ± 3.1 days for the patients who subsequently received apixaban or rivaroxaban, respectively. Meanwhile, the mean duration of the administration of the lead-in oral anticoagulant was 4.0 ± 1.1 or 15.0 ± 3.1 days for the patients who received apixaban or rivaroxaban, respectively. The overall mean duration of the combined parenteral and oral lead-in anticoagulants was 6.2 ± 0.8 or 20.7 ± 0.8 days for the patients who received apixaban or rivaroxaban, respectively. Table 4 presents a summary of the types of anticoagulants used and the duration of the lead-in therapy for both groups. 

### 3.3. Clinical Outcomes

#### 3.3.1. rVTE

Only two rVTE events occurred during hospitalization, both of which were in the recommended-lead-in group. Subsequently, two additional events were observed in the recommended-lead-in group (0.7%) during the 30 days of follow-up. Overall, the number of rVTE events within 90 days from the indexed VTE was four in four different patients in the recommended group compared to one event in the mixed group (1.4% vs. 1.4%; *p* = 1.000). Table 5 displays the results for all the outcomes.

#### 3.3.2. MB

The proportion of patients who developed MB during hospitalization was higher in the mixed group compared to the recommended-lead-in group (9.7% vs. 3.7%), but the difference was not statistically significant. In the 30 days of follow-up, only four events were reported, all of which were in the recommended-lead-in group. Overall, the total number of MB events within 90 days from the indexed VTE date included 16 events that occurred in 14 patients in the recommended group compared to eight events in seven patients in the mixed group (9.7% vs. 4.7%; *p* = 0.150). The MB events were mainly located in the abdominal cavity (three in the recommended group and one in the mixed group) and pelvic area (one in each group), with the site unknown for the remaining events. Anticoagulation held for a short time in five patients, and nine patients received a blood transfusion.

#### 3.3.3. CRNMB and Rehospitalization

Clinically relevant non-major bleeding occurred in 12 of the patients during hospitalization. This included nine in the recommended group and three in the mixed-lead-in group. During the 90 days of follow-up, 39 events occurred in 36 patients; of these events, 32 occurred in 29 patients in the recommended group compared to seven events in seven patients in the mixed group (9.8% vs. 9.7%; *p* = 0.984). The CRNMB events were mainly located in the vaginal area (six in the recommended group and one in the mixed group), epistaxis (three in the recommended group and one in the mixed group), and hematuria (two in the recommended group), with an unknown site for the remaining events. Anticoagulation took place for a short time in five patients, and one patient received a blood transfusion.

During the first 30 days of follow-up, seven patients (2.4%) from the recommended-lead-in group needed rehospitalization for VTE-related reasons. Subsequently, in the following 60 days (days 31–90 from the indexed VTE event), six additional patients (2.0%) were rehospitalized in the recommended group, compared to two in the mixed group (2.8%).

#### 3.3.4. Sub-Group Analysis

A sub-group analysis was conducted for the patients in the recommended group based on whether they had received a parenteral anticoagulant. Among these patients, 98 (33.1%) did not receive a parenteral anticoagulant, while 198 received parenteral anticoagulation (mostly enoxaparin), as indicated in Table 4. Two rVTE events occurred in patients who did not receive parenteral anticoagulation (2.0%), as well as two events in patients who received parenteral anticoagulation (1.0%) during the 90 days of follow-up. Five MB events occurred in five patients who did not receive parenteral anticoagulation (5.1%) compared to 11 events in nine patients who received parenteral anticoagulation (4.5%) during the 90 days of follow-up. Five patients who did not receive parenteral anticoagulation (5.1%), compared to 24 patients among those who received parenteral anticoagulation (12.1%), had CRNMB during the 90 days of follow-up. More rehospitalizations occurred within 90 days in patients who received parenteral anticoagulation compared to patients who did not (6.1% vs. 1.0%). Although these results mostly indicate a trend toward worsening outcomes for patients who received parenteral anticoagulants, none of these results were statistically significant, as summarized in Appendix A. 

## 4. Discussion

The current study compared the effectiveness and safety of mixed-lead-in dosing using the subtracting strategy to that of the recommended full-lead-in dosing of apixaban and rivaroxaban in patients with acute VTE. No statistically significant differences were observed in any of the effectiveness or safety outcomes between the two treatment regimens during hospitalization or within the 90 days of follow-up. Although the initial higher rate of rVTE during hospitalization in the recommended-lead-in group could be attributed to higher rates of historical VTE events or recent major surgery in that group, this rate was eventually matched by the later events that occurred in the mixed group. Overall, a trend toward a higher risk of MB was observed in the mixed group compared to the recommended group. Conversely, the recommended-lead-in group had higher rates of rehospitalization for VTE-related causes. That said, the overall study findings suggest that the use of the mixed-lead-in regimen while applying the subtraction strategy can be considered an option for patients undergoing acute VTE treatment. 

The trends in the demographic and clinical characteristics of our patients indicate that clinicians tend to follow the labeled recommended-lead-in regimen in younger patients at low risk of bleeding who present with more severe forms of VTE events. This trend is similar to findings reported by Williams et al., who observed that the patients in the reduced-lead-in group in their study were significantly older and sicker than the patients in the full-lead-in group [12]. Although the mixed-lead-in regimen was followed in about 20% of the patients in this study, similar to Williams et al.’s study (22%), this low rate of mixed-lead-in utilization should be interpreted in light of the patients included in and excluded from the study. Most of the patients included in our study were relatively young, with a low-to-intermediate risk of bleeding and normal kidney function. By contrast, many of the excluded patients had used parenteral anticoagulants for a longer duration than the maximum allowed during this study, most likely because these patients were either sicker or more fragile than the patients included in this study. With this in mind, maximum and minimum durations of anticoagulant therapy were chosen to allow for reasonable within- and between-group comparisons.

Although our study did not find any significant difference in the incidence of rVTE with comparable rates between the two groups, Williams et al. observed a trend toward a higher risk of rVTE in the reduced-lead-in group (5% vs. 1%; *p* = 0.205). Notably, the patients who received no lead-in doses were combined into the reduced-lead-in group of Williams et al.’s study, making that study’s within- and between-group comparisons different from ours [12]. The trend toward a higher risk of MB that we observed in the mixed group compared to the recommended-lead-in group was also noted by Williams et al. when comparing full- to reduced-lead-in therapy. However, this variation in the rates of MB events could be explained by the higher proportion of patients with historical MB events, thrombophilia, or active cancer in the mixed-lead-in group, as well as the significantly lower baseline Hgb level in our study. Likewise, the trend observed in Williams et al.’s study can be explained by the differences in baseline characteristics between the two groups [12].

The evidence about the appropriateness of the subtraction strategy, represented by the mixed-lead-in regimen in our study, was lacking because this group of patients was not part of the AMPLIFY or EINSTEIN trials [7,8,9]. However, the mean duration of the parenteral anticoagulants used in the recommended group was comparable to that reported in the landmark trials. In comparison to the landmark trials of apixaban and rivaroxaban, we observed lower rates of rVTE in our cohort, at 1.4% compared to 2.3% with apixaban and about 2.1% with rivaroxaban, acknowledging that we limited our follow-up to 90 days from the indexed VTE event [7,8,9]. In addition, we observed a higher rate of MB (6.5%) compared to 0.6% with apixaban and about 1.0% with rivaroxaban in the clinical trials. The subgroup analysis revealed slightly lower rates of rVTE in the group that received parenteral anticoagulation but increased rates of MB, CRNMB, and rehospitalization; these findings suggest the suitability of omitting the initial parenteral therapy.

The present study has some limitations, which include its retrospective nature and the small number of patients, especially because the limited size of the mixed-lead-in group did not allow for any further sub-analysis. The patients included were relatively young, with a low risk of bleeding and good kidney function, limiting the generalization of our findings to older, more fragile patients or patients with poor kidney function. The bleeding-risk-assessment tool used also introduced some limitations, since it only considers the patient’s age, sex, and history of cancer. That said, no specific bleeding-risk-assessment tool is currently recommended by the clinical guidelines [15,16,17,18]. Moreover, the duration of lead-in therapy was determined using discharge-medication information for our patients. Thus, the patients’ compliance for the entire duration of the lead-in therapy could not be assessed or confirmed. Due to the need to rely on data documented in the follow-up notes for the patients, we could have missed some events that were not documented in the patient files.

## 5. Conclusions

The lead-in therapy with the recommended or mixed regimens showed comparable effectiveness and safety outcomes. Subtracting parenteral-anticoagulation days from the total lead-in regimen for both apixaban and rivaroxaban might be a reasonable strategy. However, large and prospective studies are required to verify these findings.

## Figures and Tables

**Figure 1 jcm-12-00199-f001:**
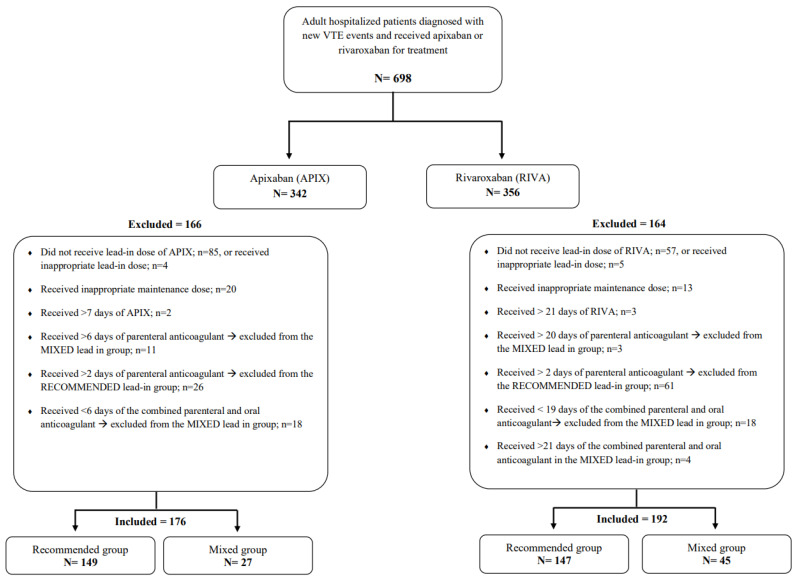
Flow diagram of included and excluded patients in the study. Abbreviations: VTE: venous thromboembolism; APIX: apixaban; RIVA: rivaroxaban.

**Table 1 jcm-12-00199-t001:** Patient demographics, clinical characteristics, and risk factors for VTE recurrence.

Patient Characteristics	Overall	Lead-in Group		*p*-Value
		Mixed	Recommended	
Overall number of patients	368	72	296	
Age in years, mean (SD)	53.1 ± 19.6	55.2 ± 19.2	52.6 ± 19.7	0.316
BMI (kg/m^2^), mean (SD)	30.8 ± 7.0	29.3 ± 7.1	31.1 ± 6.9	0.054
Hospital length of Stay (days)	6.1 ± 13.3	12.2 ± 25.3	4.6 ± 7.4	<0.0001
Gender				0.355
Male	131 (35.6)	29 (40.3)	102 (34.5)	
Female	237 (64.4)	43 (59.7)	194 (65.5)	
Pre-existing conditions				
Atrial fibrillation	8 (2.2)	3 (4.2)	5 (1.7)	0.192
Coronary artery disease	19 (5.2)	6 (8.3)	13 (4.4)	0.229
Hypertension	137 (37.2)	33 (45.8)	104 (35.1)	0.092
Valvular disease	3 (0.8)	1 (1.4)	2 (0.7)	0.481
Stroke	33 (9.0)	5 (6.9)	28 (9.5)	0.503
Transient ischemic attack	3 (0.8)	0 (0.0)	3 (1.0)	1.000
Diabetes mellitus	124 (33.7)	27 (37.5)	97 (32.8)	0.446
Chronic kidney disease	18 (4.9)	3 (4.2)	15 (5.1)	1.000
Active smoking	28 (7.6)	5 (6.9)	23 (7.8)	0.323
Thrombophilia	13 (3.5)	3 (4.2)	10 (3.4)	0.419
Active cancer	12 (3.3)	5 (6.9)	7 (2.4)	0.061
On chemotherapy (among patients with cancer)	5 (41.7)	2 (40.0)	3 (42.9)	1.000
History of MB (within 12 months)	22 (6.0)	8 (11.1)	14 (4.7)	0.058
History of CRNMB (within 12 months)	11 (3.0)	1 (1.4)	10 (3.4)	0.698
History of any bleeding (within 12 months)	8 (2.2)	2 (2.8)	6 (2.0)	0.651
Concomitant antithrombotic medications				
Aspirin	54 (14.7)	9 (12.5)	45 (15.2)	0.547
P2Y12 Inhibitors	10 (2.7)	3 (4.2)	7 (2.4)	0.623
Risk Factors for VTE recurrence				
History of previous VTE	39 (10.6)	4 (5.6)	35 (11.8)	0.119
Type of historical VTE				0.0006
DVT	27 (69.2)	0 (0.0)	27 (0.8)	---
PE	9 (23.1)	4 (100)	5 (14.3)	0.638
DVT plus PE	3 (7.7)	0 (0.0)	3 (8.6)	
Time of historical VTE				0.258
Within 3 months	3 (7.7)	0 (0.0)	3 (8.6)	
Within 12 months	2 (5.1)	0 (0.0)	2 (5.7)	
Within >12 months	32 (82.1)	3 (75.0)	29 (82.9)	
Use of oral contraceptive or ERT	43 (11.7)	5 (6.9)	38 (12.8)	0.148
Obesity (BMI ≥ 30)	181 (49.2)	31 (43.1)	150 (50.7)	0.349
Immobility	83 (22.6)	18 (25.0)	65 (22.0)	0.515
Major general surgery (within one year)	41 (11.1)	11 (15.3)	30 (10.1)	0.254
Time of major surgery				0.550
Within 3 months	30 (73.2)	7 (63.6)	23 (76.7)	
Within 3–6 months	3 (7.3)	2 (18.2)	1 (3.3)	
Within 6–12 months	3 (7.3)	1 (9.1)	2 (6.7)	
Within >12 months	4 (9.8)	1 (9.1)	3 (10.0)	
Orthopedic surgery (within one year)	29 (7.9)	8 (11.1)	21 (7.1)	0.282
Time of orthopedic surgery				0.622
Within 3 months	25 (86.2)	8 (100.0)	17 (81.0)	
Within 12 months	2 (6.9)	0 (0.0)	2 (9.5)	
Within >12 months	1 (3.4)	0 (0.0)	2 (9.5)	

Results are presented as frequency (percentage) or mean ±SD. The *p*-values are from the *t*-test for continuous data and chi-square or Fisher’s exact test for categorical data. Abbreviations: BMI: body mass index; VTE: venous thromboembolism; DVT: deep-vein thrombosis; PE: pulmonary embolism; ERT: estrogen-replacement therapy; MB: major bleeding; CRNMB: clinically relevant non-major bleeding; SD: standard deviation.

**Table 2 jcm-12-00199-t002:** Laboratory values at the lead-in dose initiation.

Laboratory Values	Overall	Lead-in Group		*p*-Value
		Mixed	Recommended	
Scr at lead-in dose initiation (mg/dL)	0.8 ± 0.3	0.71 ± 0.2	0.77 ± 0.3	0.120
eGFR at lead-in dose initiation (mL/min/1.73 m^2^)	102.7 ± 46.8	115.4 ± 65.7	99.6 ± 40.5	0.009
CrCl at lead-in dose initiation (ml/min)	103.2 ± 50.0	109.7 ± 68.3	101.7 ± 44.5	0.223
Hgb level before lead-in dose initiation (g/dL)	12.5 ± 2.1	11.7 ± 2.2	12.7 ± 2.0	<0.001

Results are presented as mean ±SD. The *p*-values are from the *t*-test. Abbreviations: Scr: serum creatinine; eGFR: estimated glomerular filtration rate; CrCl: creatinine clearance; Hgb: hemoglobin; SD: standard deviation.

**Table 3 jcm-12-00199-t003:** Type and etiology of the new VTE event and the risk of bleeding.

Patient Characteristics	Overall	Lead-in Group		*p*-Value
		Mixed	Recommended	
Overall number of patients	368	72	296	
Type of current VTE event				<0.001
DVT	151 (41.0)	10 (13.9)	141 (47.6)	0.601
Proximal	109 (72.2)	8 (80.0)	101 (71.6)	
Distal	14 (9.3)	0 (0.0)	14 (9.9)	
Mixed	21 (13.9)	2 (20.0)	19 (13.5)	
Unspecified	7 (4.6)	0 (0.0)	7 (5.0)	
PE	188 (51.1)	54 (75.0)	134 (45.3)	0.086
Segmental	68 (36.2)	16 (29.6)	52 (38.8)	
Subsegmental	28 (14.9)	5 (9.3)	23 (17.2)	
Mixed	58 (30.9)	18 (33.3)	40 (29.9)	
Unspecified	34 (18.1)	15 (27.8)	19 (14.2)	
DVT plus PE	29 (7.9)	8 (11.1)	21 (7.1)	
DVT type				0.562
Proximal	12 (41.4)	3 (37.5)	9 (42.9)	
Distal	6 (20.7)	1 (12.5)	5 (23.8)	
Mixed	5 (17.2)	1 (12.5)	4 (19.0)	
Unspecified	6 (20.7)	3 (37.5)	3 (14.3)	
PE type				0.468
Segmental	11 (37.9)	3 (37.5)	8 (38.1)	
Subsegmental	2 (6.9)	1 (12.5)	1 (4.8)	
Mixed	9 (31.1)	1 (12.5)	8 (38.1)	
Unspecified	7 (24.1)	3 (37.5)	4 (19.0)	
VTE Etiology				0.694
Provoked	203 (55.2)	37 (51.4)	166 (56.1)	
Unprovoked	81 (22.0)	16 (22.2)	65 (22.0)	
Not reported	84 (22.8)	19 (26.4)	65 (22.0)	
Bleeding Risk ^a^				0.297
High risk	11 (3.0)	4 (5.6)	7 (2.4)	
Intermediate risk	283 (76.9)	52 (72.2)	231 (78.0)	
Low risk	74 (20.1)	16 (22.2)	58 (19.6)	

Results are presented as frequency (percentage). The *p*-values are from the chi-square or Fisher’s exact test. ^a^ From the bleeding-risk-assessment score [13]. Abbreviations: VTE: venous thromboembolism; DVT: deep-vein thrombosis; PE: pulmonary embolism.

**Table 4 jcm-12-00199-t004:** Recommended vs. mixed-lead-in dosing for VTE treatment.

Patient Status	Overall	Apixaban	Rivaroxaban
Overall number of patients	368	176	192
Recommended-lead-in dosing group	296	149	147
Type of parenteral anticoagulant used			
LMWH (enoxaparin)	168 (56.8)	68 (45.6)	100 (68.0)
UFH	29 (9.8)	22 (14.8)	7 (4.8)
Fondaparinux	1 (0.3)	1 (0.7)	0 (0.00)
None	98 (33.1)	58 (38.9)	40 (27.2)
Duration for the received lead-in dose of DOAC (days)		7.0 ± 0.0	21.0 ± 0.0
Duration for the received parenteral anticoagulant (days)		1.3 ± 0.4	1.2 ± 0.4
Duration for the received parenteral anticoagulant and DOAC (days)		8.3 ± 0.4	22.2 ± 0.4
Mixed-lead-in dosing group	72	27	45
Type of parenteral anticoagulant used			
LMWH (enoxaparin)	61 (84.7)	21 (77.8)	40 (88.9)
UFH	11 (15.3)	6 (22.2)	5 (11.1)
Duration of the received lead-in dose of DOAC (days)		4.0 ± 1.1	15.0 ± 3.1
Duration of the received parenteral anticoagulant (days)		2.2 ± 1.4	5.6 ± 3.1
Duration of the received parenteral anticoagulant and DOAC (days)		6.2 ± 0.8	20.7 ± 0.8

Results are presented as frequency (percentage) or mean ± SD. Abbreviations: VTE: venous thromboembolism; LMWH: low-molecular-weight heparin; UFH: unfractionated heparin; DOAC: direct oral anticoagulant; SD: standard deviation.

**Table 5 jcm-12-00199-t005:** Outcomes during hospitalization and up to 90 days after the VTE event.

Patient Characteristic	Overall	Lead-in Group		*p*-Value
		Mixed	Recommended	
Overall number of patients	368	72	296	
rVTE event				
During hospitalization	2 (0.5)	0 (0.0)	2 (0.7)	1.000
Within 30 days ^a^	2 (0.5)	0 (0.0)	2 (0.7)	1.000
Cumulative within 90 days ^b^	3 (0.8)	1 (1.4)	2 (0.7)	0.481
Patients with at least one rVTE within 90 days ^c^	5 (1.4)	1 (1.4)	4 (1.4)	1.000
MB event				
During hospitalization	18 (4.9)	7 (9.7)	11 (3.7)	0.060
Within 30 days	4 (1.1)	0 (0.0)	4 (1.4)	1.000
Cumulative within 90 days	6 (1.6)	1 (1.4)	5 (1.7)	1.000
Patients with at least one MB within 90 days	21 (5.7)	7 (9.7)	14 (4.7)	0.150
CRNMB event				
During hospitalization	12 (3.3)	3 (4.2)	9 (3.0)	0.890
Within 30 days	27 (7.3)	4 (5.6)	23 (7.8)	0.518
Cumulative within 90 days	27 (7.3)	4 (5.6)	23 (7.8)	0.518
Patients with at least one CRNMB within 90 days	36 (9.8)	7 (9.7)	29 (9.8)	0.984
Rehospitalization ^d^				
Within 30 days	7 (1.9)	0 (0.0)	7 (2.4)	0.353
Cumulative within 90 days	15 (4.1)	2 (2.8)	13 (4.4)	0.745
Death during hospitalization	0 (0.0)	0 (0.0)	0 (0.0)	---

Results are presented as frequency (percentage). The *p*-values are from the chi-square or Fisher’s exact test. ^a^ Within 30 days: 30 days from the indexed event date (excluding hospitalization days). ^b^ Within 90 days: 90 days from the indexed event date (excluding hospitalization days). ^c^ Cumulative number of patients with the outcome within 90 days, including events occurring during index hospitalization. ^d^ Rehospitalization due to VTE-related causes (recurrence, deterioration, or bleeding). Note: Two patients in the recommended group and one in the mixed-lead-in group had two MB events each, and three patients from the recommended group each had two CRNMB events each during the 90 days of follow-up after the indexed VTE event, including events that occurred during the index hospitalization. Abbreviations: rVTE: recurrent venous thromboembolism; MB: major bleeding; CRNMB: clinically relevant non-major bleeding.

## Data Availability

All data are presented in this article.

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
