# Peer review of "Comparative Effectiveness of Apixaban and Rivaroxaban Lead-in Dosing in VTE Treatment: Observational Multicenter Real-World Study"

_jcm, 2022, doi:10.3390/jcm12010199_

Round 1
Reviewer 1 Report (Previous Reviewer 1)
I continue to consider the usefulness of studies like the present one , especially for medicines whose activity does not require a hemostatic monitoring. It is the best modality for finding the adequate therapy regimen.
Considering the probable inaccuracies of some "p" values from table 1, I would suggest a request for consultation of a specialist in medical statistics.
Author Response
We appreciate the feedback from reviewer 1, and we have revised the p-values in the resubmitted manuscript (highlighted in yellow). Also, a mistake was noticed in the categories for the period of orthopedic surgery, which has been corrected and highlighted in yellow.
Reviewer 2 Report (New Reviewer)
Dear authors
thank you for this interesting article
please describe the type of major bleeding complications or clinically relevant and type of (neuro)surgical intervention/outcome.
What was the "antagonisation" protocol for emergency neurosurgical interventions?
Thank you!
Author Response
We appreciate the feedback from reviewer 2, and we have collected the necessary information to describe the bleeding complications from all participating centers. We have added two short descriptions under the results section for both MB and CRNMB that describe the bleeding complications (highlighted in yellow). None of the patients who experienced bleeding complications received any reversal agents or surgical interventions.
This manuscript is a resubmission of an earlier submission. The following is a list of the peer review reports and author responses from that submission.
Round 1
Reviewer 1 Report
It is a valuable study focusing on the most efficacious therapeutical approach to overcome the critical high-risk time frame post-VTE. Aware of the AMPLIFY, EINSTEIN, HOKUSAI -VTE, RE -COVER trials and M. Williams work the authors are presenting two modalities for optimization of anticoagulant therapy used post-VTE: oral anticoagulants (apixaban and rivaroxaban) in two dosing strategies in mixed lead- in and in recommended regimen, respectively associated with parenteral anticoagulants (LMWH, UFH, Fondaparinux ). Using these methods they conducted a multicenter retrospective study, selecting on rigorous criteria 368 patients from a cohort of 698 adult hospitalized patients diagnosed with new VTE events. A comprehensive evaluation has been performed in light of the scarcity of robust evidence in this field. They aimed to assess the effectiveness and safety of the two lead-in regimens on the incidence of recurrent VTE, and significant and clinically relevant non-major bleeding in patients with acute VTE events. Based on descriptive statistical assessments, the authors concluded that the tested two regimens have comparable effectiveness and safety outcomes, underlining that subtracting parenteral anticoagulation days from the total lead-in regimen might be for both apixaban and rivaroxaban a reasonable strategy
Limitations:
-a small number of the studied cohort, the limited size of the mixed lead-in group, and above all of the number of patients with long list of different characteristics; they are connected with limitations of the statistical evaluations
-the uncertainty of the patient's compliance for the entire duration of the lead-in therapy in case of using discharge medication information from the patients
Corrections:
-complete in table 4 the duration with days
- please revise table 1 regarding the p values
-reference 12- not complete